# Investigating the Mitoprotective Effects of S1P Receptor Modulators Ex Vivo Using a Novel Semi-Automated Live Imaging Set-Up

**DOI:** 10.3390/ijms25010261

**Published:** 2023-12-23

**Authors:** Rebecca Ludwig, Bimala Malla, Maria Höhrhan, Carmen Infante-Duarte, Lina Anderhalten

**Affiliations:** 1Experimental and Clinical Research Center (ECRC), 13125 Berlin, Germany; rebecca.ludwig@charite.de (R.L.); lina-carlotta.anderhalten@charite.de (L.A.); 2Charité—Universitätsmedizin Berlin, 10117 Berlin, Germany; 3Max-Delbrück-Center for Molecular Medicine (MDC) in the Helmholtz Association, 13125 Berlin, Germany; 4Institute for Medical Immunology, 13353 Berlin, Germany

**Keywords:** multiple sclerosis, neurodegeneration, chronic organotypic hippocampal slice culture, oxidative stress, sphingosine-1-phosphate receptor modulator, siponimod, fingolimod, neuronal mitochondrial dynamics, live imaging, confocal fluorescence microscopy

## Abstract

In multiple sclerosis (MS), mitochondrial alterations appear to contribute to disease progression. The sphingosine-1-phosphate receptor modulator siponimod is approved for treating secondary progressive MS. Its preceding compound fingolimod was shown to prevent oxidative stress-induced alterations in mitochondrial morphology. Here, we assessed the effects of siponimod, compared to fingolimod, on neuronal mitochondria in oxidatively stressed hippocampal slices. We have also advanced the model of chronic organotypic hippocampal slices for live imaging, enabling semi-automated monitoring of mitochondrial alterations. The slices were prepared from *B6.Cg-Tg(Thy1-CFP/COX8A)S2Lich/J* mice that display fluorescent neuronal mitochondria. They were treated with hydrogen peroxide (oxidative stress paradigm) ± 1 nM siponimod or fingolimod for 24 h. Afterwards, mitochondrial dynamics were investigated. Under oxidative stress, the fraction of motile mitochondria decreased and mitochondria were shorter, smaller, and covered smaller distances. Siponimod partly prevented oxidatively induced alterations in mitochondrial morphology; for fingolimod, a similar trend was observed. Siponimod reduced the decrease in mitochondrial track displacement, while both compounds significantly increased track speed and preserved motility. The novel established imaging and analysis tools are suitable for assessing the dynamics of neuronal mitochondria ex vivo. Using these approaches, we showed that siponimod at 1 nM partially prevented oxidatively induced mitochondrial alterations in chronic brain slices.

## 1. Introduction

Multiple sclerosis (MS) is an autoimmune, inflammatory, demyelinating, and neurodegenerative disease of the central nervous system (CNS) [1]. It is a major cause of chronic neurological disability in young adults and affects approximately 2.8 million people worldwide [1]. Usually, the disease presents with initial symptoms in individuals between 20 and 50 years of age [2]. Clinical symptoms include optic neuritis, paresthesia, paresis, spasticity, bladder and bowel dysfunction, and cognitive impairment [1,3]. The majority of patients are initially diagnosed with relapsing-remitting MS (RRMS) and suffer from relapses due to new inflammatory CNS lesions, followed by a period of total or partial remission [4,5,6]. Of these patients, around two-thirds develop secondary progressive MS (SPMS), which is characterized by increasing neurological impairment without phases of clear relapses [7]. Approximately 10% of patients are diagnosed with primary progressive MS, experiencing advancing disabilities from the onset of the disease [4]. While inflammation by CNS-invading peripheral immune cells is believed to contribute primarily to the RR phase, other pathophysiological mechanisms, especially inflammation-induced neurodegeneration, are thought to predominate in PPMS and SPMS and drive disease progression [6,8,9].

Mitochondrial dysfunction has emerged as a potential key player in MS that affects neurons with intact myelin sheaths, thereby contributing to axonal loss, even during the early stages of the disease [10,11,12,13]. One possible link between neuroinflammatory and neurodegenerative aspects of MS is the involvement of inflammatory mediators, such as reactive oxygen and nitrogen species (ROS and RNS) produced by activated microglia as well as CNS-invading immune cells [14]. In a physiological state, mitochondrial ATP production results in the creation of low ROS levels, which are efficiently countered by the cellular antioxidant systems [15]. Mitochondrial dysfunction itself leads to increased production of mitochondrial ROS during oxidative phosphorylation, causing a vicious cycle of ROS-induced ROS production [16,17]. In this context, ROS can trigger mitochondrial alterations, one of the earliest events of focal neurodegeneration [13,18]. Thus, mitochondria may represent a possible target for anti-neurodegenerative therapeutics.

Most of the disease-modifying therapies (DMTs) approved for RRMS aim to reduce the number of relapses and prevent or postpone the transition to SPMS, although most of them, so far, had little impact on disability accumulation [19]. Active or relapsing SPMS can be targeted with different specific DMTs, including siponimod, ponesimod, ofatumumab, ocrelizumab, interferon beta 1a and 1b s.c., cladribine, and mitoxantrone [20,21,22,23]. However, it is to be noted that guidelines and recommendations for the treatment of SPMS still undergo constant changes and differ between the European Union and the United States [23]. For PPMS, only ocrelizumab, a humanized monoclonal antibody targeting CD20-positive cells, has been approved up to date [24,25]. Very recent evidence implies that MS severity is substantially associated with genetic variants related to CNS resilience genes [26]. There remains an unmet requirement for targeted treatment options that address the neurodegenerative aspects of the disease.

Fingolimod is a first-generation sphingosine-1-phosphate receptor (S1PR) modulator, targeting mostly S1PR1, 3, and 5 [27]. It is an oral DMT that has been approved for RRMS [28]. Fingolimod induces lymphocyte homing and modulates lymphocyte egress from lymph nodes, effectively reducing the number of circulating lymphocytes and their infiltration to sites of CNS inflammation [29,30]. In neuronal cell cultures, fingolimod was demonstrated to prevent oxidative stress-induced alterations in mitochondrial morphology and to restore mitochondrial function [31,32].

Siponimod (BAF312) is a next-generation S1PR modulator approved for SPMS that specifically targets S1PR1 and 5 [20,33,34]. These particular receptors are involved in the regulation of inflammatory responses and in myelination [33,35]. Importantly, siponimod does not modulate S1PR3, which is located on cardiac cells and is thought to be mainly responsible for cardiac complications, such as bradycardia and arrhythmia. These cardiac complications are observed more commonly with fingolimod but less frequently for siponimod [36,37]. In patients with SPMS, oral treatment with siponimod showed both anti-inflammatory and neuroprotective effects [20,35] and appears to contribute to a reduction in disability progression [20]. The protective effects of siponimod may result from processes such as attenuation of microglial activation, suppression of pro-inflammatory pathways, alteration of action potential firing, as well as modulation of hyperpolarization-activated and cyclic nucleotide-gated ion channels [37,38,39,40,41,42]. Siponimod was also reported to effectively suppress experimental autoimmune encephalitis (EAE), a murine model of MS, and to partially restore cortical neuronal circuit function in combined general and focal EAE [43].

We hypothesized that siponimod’s neuroprotective effects involve mitochondrial pathways. In this study, we aimed to investigate the effect of siponimod on neuronal mitochondrial morphology and motility in oxidatively stressed chronic organotypic hippocampal slice cultures ex vivo and to compare its influence with the effects of the preceding compound, fingolimod. To assess mitochondrial dynamics, we have developed a mitochondrial imaging approach and assessed middle- to long-term neuromitoprotective substance effects using this ex vivo model. To evaluate mitochondrial alterations in a standardized manner, a semi-automated analysis tool was established in the context of this study.

## 2. Results

### 2.1. Establishment of a Semi-Automated Analysis of Mitochondrial Motility and Morphology

Here, we established a live imaging protocol with chronic hippocampal slice cultures that allows the standardized monitoring of neuronal mitochondrial alterations and assessment of middle- to long-term treatment responses using confocal fluorescence microscopy. An in-house developed macro for ImageJ software (version 1.53v; for full macro, see Appendix A) enabled us to assess the mitochondrial morphology and motility in live imaging sequences in a semi-automated manner. The following paragraph briefly offers an overview of the analysis steps implemented in the ImageJ macro (Figure 1). To track mitochondrial motility, each sequence was processed separately. First, images were corrected for slice movement in the X- and Y-directions using the StackReg plugin in ImageJ (https://bigwww.epfl.ch/thevenaz/stackreg/, accessed on 27 December 2022) [44,45]. Thereafter, the actual mitochondrial tracking was performed using the TrackMate plugin (version 4.0.1) [45,46]. Here, the LoG detector was chosen to assess the mean mitochondrial track speed and track displacement, detecting particles with an estimated blob diameter of 1.5 μm, with a threshold of 1 μm. Further tracking parameters were set to a frame-to-frame linking of 2 μm without any gap closing. For further analysis, only track displacement values above 1.5 μm displacement were considered (Figure 1). For the morphological analysis, only the first of 31 images per sequence was considered. The threshold of particle sizes analyzed was set to exclude events below 0.2 μm^2^ and above 5 μm^2^ area (Figure 1). Once, the semi-automated image analysis was completed, relevant datasets concerning mitochondrial speed and displacement, mitochondrial area, Feret’s diameter (mitochondrial length), and aspect ratio (shape from perfect circularity (1) to elongation (>1–∞)) were extracted for statistical group comparisons. Finally, the fraction of motile mitochondria was calculated from morphology and motility measures (*n* = 12 ROIs).

### 2.2. Siponimod Prevents Oxidative Stress-Induced Alterations in Mitochondrial Morphology

To investigate the effects of siponimod on stressed neuronal mitochondria, chronic hippocampal slice cultures were generated from *mitoCFP* mouse pups and exposed to oxidative stress by hydrogen peroxide (H_2_O_2_) using the stress paradigm applied in our previous studies on spinal roots and acute brain slices [47,48,49,50]. We assessed the effect of siponimod on oxidative stress induction in neuronal mitochondria inside the dentate gyrus [51] using the imaging and analysis tools described above (detailed in Section 4) and compared it to the influence of its preceding compound, fingolimod.

The analysis of mitochondrial morphology in the DG of hippocampal slices (Figure 2A,B) revealed a significant reduction in mitochondrial length (H_2_O_2_ vs. negative control, 1.400 ± 0.353 µm vs. 1.469 ± 0.317 µm; *p* < 0.0001) and area (H_2_O_2_ vs. negative control, 0.698 ± 0.371 µm^2^ vs. 0.755 ± 0.324 µm^2^; *p* < 0.0001) in H_2_O_2_-treated tissue slices (24 h at 100µM; see Section 4) compared to negative control slices (Figure 2C,D). Siponimod treatment for 24 h was able to diminish oxidative stress-induced alterations in mitochondrial morphology. The H_2_O_2_-induced reduction in mitochondrial length was significantly lowered (H_2_O_2_ + siponimod vs. H_2_O_2_, 1.433 ± 0.362 µm vs. 1.400 ± 0.353 µm; *p* = 0.0028) (Figure 2C), and there was a trend towards larger mitochondrial area in slices treated with H_2_O_2_ + siponimod compared to H_2_O_2_ only (Figure 2D). The treatment with fingolimod did not lead to significant prevention of H_2_O_2_-induced morphological alterations; however, we observed a trend for longer and larger mitochondria compared to positive control slices (Figure 2C,D). The mitochondrial aspect ratio was not affected by any tested treatment conditions compared to negative controls (Figure 2E). Table 1 provides a summary of results on mitochondrial morphology. In addition, the Appendix A shows the outline masks generated for morphological analysis.

### 2.3. Siponimod Reduces Oxidative Stress-Induced Changes in Mitochondrial Motility

In oxidatively stressed hippocampal slices, mitochondria traveled slightly shorter distances compared to negative controls; however, this effect was not significant (Figure 3B). Upon slice treatment with H_2_O_2_, we observed no effect on mitochondrial track speed compared to negative control slices (Figure 3C). However, we observed a decrease in the fraction of motile mitochondria in H_2_O_2_-treated slices compared to negative controls (H_2_O_2_ vs. negative control, 0.220 ± 0.055 vs. 0.277 ± 0.128) (Figure 3A,D). Treatment with siponimod showed a trend to prevent the oxidative stress-induced decrease in mitochondrial displacement (Figure 3B). Moreover, siponimod treatment (H_2_O_2_ + siponimod vs. H_2_O_2_, 0.244 ± 0.081 µm/s vs. 0.219 ± 0.081 µm/s; *p* < 0.0001) and, in particular, fingolimod treatment (H_2_O_2_ + fingolimod vs. H_2_O_2_, 0.296 ± 0.085 µm/s vs. 0.219 ± 0.081 µm/s; *p* < 0.0001) led to a significant increase in track speed compared to H_2_O_2_ treatment alone (Figure 3C). Additionally, in slices treated with siponimod or fingolimod, the fractions of motile mitochondria approached that of control slices (Figure 3A,D). Due to the low sample size of computable fractions (*n* = 12), this effect was not significant, but H_2_O_2_ + fingolimod reached borderline significance when compared to H_2_O_2_ alone (H_2_O_2_ + fingolimod vs. H_2_O_2_, 0.269 ± 0.087 vs. 0.220 ± 0.054; *p* = 0.096). This effect was also observable by visual inspection, as shown in Figure 3A, while morphological changes became mainly visible through semi-automated analysis (Figure 2C,D; Appendix A). Table 2 provides a summary of results on mitochondrial motility.

## 3. Discussion

In this study, we present an ex vivo model of chronic organotypic hippocampal slices to monitor mitochondrial dynamics in neurons and drug effects upon these properties over days. Our group works on both, animal models of MS, namely EAE, and also on different ex vivo models to mimic oxidative stress on mitochondria [48,49,52]. In this context, acute brain slices and spinal root explants represent an option to monitor mitochondrial dynamics over short time periods in living tissue ex vivo [47,48]. Organotypic hippocampal slice cultures preserve the three-dimensional parenchymal architecture and function [53] and responsiveness to inflammatory stimuli [54], enabling controlled substance testing without the need for in vivo application. In contrast to acute brain slices, chronic ex vivo approaches allow addressing middle- to long-term treatment effects [52] and thus might be more appropriate to predict alterations in chronic neurodegenerative paradigms, such as secondary progressive MS.

Others have examined mitochondrial respiration as an indicator of mitochondrial functionality and metabolic activity, but have used tissue homogenates of chronic organotypic hippocampal slices. [55]. To our knowledge, this is the first study using intact chronic hippocampal slices to monitor mitochondrial alterations in morphology and motility within living brain tissue and to investigate mitoprotective drug effects. In-house, we have developed a macro for ImageJ, enabling us to perform a semi-automated analysis of mitochondrial dynamics in this chronic ex vivo model. The utilization of the macro encompasses substantial advantages over manual analyses. It automates repetitive tasks and significantly facilitates and accelerates the analysis of thousands of mitochondria within the ROIs. Furthermore, the provision of a semi-automated analytical tool markedly diminishes potential analytical biases and enhances analysis consistency and precision as well as the potential for objective interpretation.

Under oxidative stress, induced by application of H_2_O_2_, we observed smaller and shorter mitochondria, which goes in line with previous morphological studies on oxidatively stressed mitochondria in acute brain slices and spinal root explants [47,48,49]. Mitochondrial morphology is determined by a physiological cycle of mitochondrial fission, also referred to as fragmentation, and fusion, allowing sufficient ATP distribution, adaption to energy demand, and quality maintenance. Thus, mitochondrial shape reflects various functional processes, such as ATP production, mitochondrial DNA management, calcium homeostasis, and the elimination of damaged mitochondrial fragments via mitophagy [56,57]. Mitochondrial fission is known to take place in stressed cells as part of damage control [58]. An imbalance in mitochondrial dynamics plays a crucial role in neurodegeneration, arising through, for example, disrupted mitochondrial transport and excessive mitochondrial fission [59]. Chronic oxidative stress is also known to promote apoptosis-related fragmentation [60]. We suppose the shorter and smaller mitochondria to be a result of oxidative stress-induced mitochondrial fission in the chronic hippocampal slices, possibly to enhance the number of functional mitochondria by getting rid of damaged membrane parts.

Siponimod reduced the oxidative stress-induced decrease in mitochondrial area and significantly prevented the observed decrease in mitochondrial length. A similar, but not statistically significant trend was observed for fingolimod, which agrees with the previous finding that it restored oxidative stress-induced morphological changes in vitro [31,32]. CNS resident cells, including oligodendrocytes, microglia, astrocytes, and neurons, express S1PR, implying potential direct neuroprotective effects of S1PR modulators [32,35].

An EAE study has suggested that siponimod may directly promote the survival of neurons as they express S1PR1 and S1PR5 [61]. Other two EAE studies using fingolimod have revealed that the pharmacological antagonism of S1PR1, expressed on astrocytes, leads to a reduction in demyelination, axonal loss, and astrogliosis [62], or ameliorates EAE progression and reduces pro-inflammatory cytokine production of innate CNS immune cells [63], respectively. Thus, although our present study focuses on the suitability of the chronic hippocampal slice model to investigate neuronal mitochondria, the model could serve as a valuable tool to explore drug effects not only on neurons but also on astrocytes and microglia since they are known to play a key role in progressive MS processes [64,65]. In future experiments, we plan, therefore, to expand our investigations to the glial components of the slices. These studies may include morphological as well as immunohistochemical and gene expression analyses of cell activation markers and effector cytokines, such as IL-1β or TNF-α, respectively [66,67,68]. Future studies investigating different resident CNS cell types should also contemplate S1PR modulator pre-treatment experiments to assess the preventative and protective effects of siponimod and/or fingolimod on the CNS.

Previous studies suggested that S1P may modulate mitochondrial function by reducing membrane depolarization and increasing mitochondrial calcium concentration and subsequent neuronal swelling, as well as by influencing cytochrome-c oxidase assembly and mitochondrial respiration [69,70]. Others have found that siponimod interacts with the cellular (anti-)oxidant system by, for example, restoring thiol levels and increasing the expression and activity of protective transcription factors [31]. Moreover, it seems to induce organelle fusion in other models [71]. We can only speculate why siponimod demonstrated a more pronounced effect on restoring mitochondrial morphology compared to fingolimod in our experiments, despite presumably antagonizing the same S1PR inside the CNS. Possible contributing factors may be distinct S1PR receptor affinities, interaction with different intrinsic pathways or cellular mechanisms, or varying pharmacokinetic profiles and potencies. In cultured CNS cells, S1PR modulator efficacy was shown within the 10–100 nM range [72,73,74]. A pharmacokinetic study on siponimod showed half maximal effective concentrations in a sub-nanomolar range for S1PR1 and 5 [75]. Siponimod titration should be part of future experiments to gain a clear insight into possible dose dependencies or two-sided effects of siponimod on neuronal mitochondria. How the observed neuromitoprotective effect is then transmitted on the molecular scale remains largely unclear.

Regarding mitochondrial motility, we observed that oxidatively stressed mitochondria in chronic hippocampal slices moved slightly shorter distances than mitochondria in negative control slices, while the track speed remained largely unaffected by oxidative stress induction. However, we observed an increase in mitochondrial track speed when slices were simultaneously incubated with siponimod or fingolimod. In slices treated with siponimod or fingolimod, the fractions of motile mitochondria approached that of control slices, but changes were not significant. O’Sullivan et al. have shown that siponimod induces an increase in intracellular calcium (Ca^2+^), probably transmitted via the SP1R [74]. It is known that mitochondrial and Ca^2+^ homeostasis are strongly interconnected [76,77]. In fact, it has been reported that, as a downstream event of oxidative stress, Ca^2+^ accumulates in neurons [78]. In our group, we have been able to show that inhibition of mitochondrial Ca^2+^ uptake prevents oxidative stress-induced alterations of mitochondrial motility and preserves the mitochondrial membrane potential in explanted murine spinal roots [49]. Ca^2+^ is known to be strongly involved in the regulation of mitochondrial motility and thereby might determine the altered mitochondrial track speed upon S1PR modulator treatment. Labeling intracellular and intramitochondrial calcium during future experiments could illuminate how they may exert effects on mitochondria.

To further determine whether the observed increase in motility upon siponimod and fingolimod treatment goes along with preserved mitochondrial functionality, further investigations, such as measurement of mitochondrial membrane potential with JC-1 dye, should be conducted. In addition, we have previously established methods to assess neuronal metabolic parameters, such as ATP content, oxygen consumption, as well as electrophysiology [48], and will apply them on chronic hippocampal slices to elucidate how expression patterns of proteins are associated with mitochondrial fusion, fission, and mitochondrial transport. We will assess fusion proteins (e.g., dynamin-related GTPases, optic atrophy 1, mitofusin 1 or 2), fission proteins (e.g., dynamin-related protein 1), movement- and cytoskeleton-related proteins (e.g., dynein, kinesin, β-actin, tubulin), and proteins related to mitochondrial immobility (e.g., syntaphilin) by qPCR [79,80,81].

With our study, we contribute to the 3R principles as first described by Russell and Burch in the 1950s [82]. The utilization of ex vivo models, such as the organotypic brain slice culture effectively reduces the number of experimental animals compared to in vivo models [83]. Mechanistic studies will be performed to adjust the model to repeated live imaging for mitochondrial monitoring, significantly further reducing the number of experimental mice. In addition, in vivo models of neuroinflammation can be substituted by inflammatory ex vivo paradigms, implementing an aspect of replacement. An ex vivo model to reliably test dose-dependent neuromitotoxic effects could facilitate studies on drug safety and kinetics. Moreover, it could be used as a platform of living tissue for investigating neuromitoprotective treatment effects and drug efficiency, bridging the gap between single-cell approaches and in vivo applications, and thus expanding the scope of potential applications.

Nevertheless, this study presents some limitations. To monitor mitochondria ex vivo in a long-term treatment setup, we present a semi-automated image analysis tool. Thus, some steps of the motility analysis, including thresholding, still have to be manually adjusted since every acquired image intrinsically differs in brightness and contrast due to, for example, slightly different ratios of neuronal bodies to axons. However, we believe the semi-automated analysis to be the best option since human eye resolution reaches its limits when observing the raw data. In addition, to our knowledge, no dose titration has yet been performed to assess the effects of siponimod on mitochondrial parameters, which represents another limitation of our study. Further, the applied H_2_O_2_ concentration, selected in accordance with previous slice culture experiments [48], is most likely not representative of physiological conditions. However, H_2_O_2_ levels were recognized to strongly diminish over a period of 24 h in vitro [84], which may, together with the relatively small sample size of ROIs per treatment condition (*n* = 12), be one possible explanation why the effect of H_2_O_2_ treatment on motility parameters was not statistically significant in our experimental setup. The specific concentrations of both H_2_O_2_ and S1PR modulators at the tissue depths where we conducted slice imaging remain undetermined and are presumably lower.

## 4. Materials and Methods

### 4.1. Ethics Statement and Animals

This study was performed in strict accordance with the European Communities Council Directive of 22 September 2010 (2010/63/EU). All experimental procedures were approved by the local authority on animal experiments in Berlin (Berlin State Office for Health and Social Affairs, LaGeSo; approval-ID: TCH0008/20). The assessment of mitochondrial alterations was conducted in the hippocampal tissue of the mouse strain *B6.Cg-Tg(Thy1-CFP/COX8A)S2Lich/J* (‘*mitoCFP*’; FEM animal facility, Berlin Buch, Germany). MitoCFP mice express a cyan fluorescent protein (CFP) coupled to a human cytochrome c oxidase subunit 8A under the control of the Thy1 promoter gene specifically in neuronal mitochondria [85], enabling the investigation of alterations in mitochondrial morphology and motility using confocal fluorescence microscopy.

### 4.2. Generation of Organotypic Hippocampal Slice Cultures

We prepared chronic organotypic hippocampal slices, as described previously [52], with adjustments to the mouse strain and imaging requirements. The steps are detailed in the Appendix A. In brief, *mitoCFP* pups aged 7 to 10 days were sacrificed by decapitation. The scalp was removed and the skull was placed in pre-cooled cutting medium (1× MEM (Gibco^TM^, Thermo Fisher Scientific Inc., Waltham, MA, USA) with 1% L-Glutamine (Merck, Darmstadt, Germany)). Under sterile conditions, the skull was opened and the brain was transferred to a fresh cutting medium. The hippocampi were isolated and sectioned into 350 µm thick slices using a McIlwain Tissue Chopper (Mickle Laboratory Engineering Co., Ltd., Guildford, Surrey, UK). Intact slices were identified under a basic light microscope and distributed across polytetrafluorethylene (PTFE) membrane culture inserts (PICM0RG50, Millicell cell culture inserts, 0.4 µm, Merck, Darmstadt, Germany) in 6-well plates in a randomized manner. The whole process of slice culture generation consistently adhered to a time frame of 30 min and two replicates per culture condition were prepared. Slices were cultured in a modified version of the culture medium used by Wang et al. [86] (1× MEM, 25% HBSS (Gibco^TM^, Thermo Fisher Scientific Inc., Waltham, MA, USA), 25% heat-inactivated horse serum (Gibco^TM^, Thermo Fisher Scientific Inc., Waltham, MA, USA), 13 mM HEPES (Gibco^TM^, Thermo Fisher Scientific Inc., Waltham, MA, USA), 35 mM glucose (B.Braun, Melsungen, Germany), Pen/Strep (Gibco^TM^, Thermo Fisher Scientific Inc., Waltham, MA, USA)) at 37 °C and 5% CO_2_, and medium was changed every two days for 14 days.

### 4.3. Experimental Ex Vivo Setup

On day 14 of the organotypic hippocampal slice culture, the substance treatment was conducted at 37 °C and 5% CO_2_ for the duration of 24 h (Figure 4). For induction of oxidative stress, slices were incubated with 100 µM H_2_O_2_ (positive control condition; Sigma-Aldrich, St. Louis, MO, USA) [87]. The selection of this specific concentration was based on previous ex vivo studies on mitochondrial dynamics [47,48,49,88], while the time of incubation was adapted to our long-term treatment approach. To assess the ability of the S1PR modulators to antagonize the effect of H_2_O_2_ on mitochondrial dynamics, slices were treated with 100 µM H_2_O_2_ + 1 nM siponimod (BAF312; Hölzel Diagnostika Handels GmbH, Cologne, Germany) or 100 µM H_2_O_2_ + 1 nM fingolimod (FTY20; Sigma-Aldrich Chemie GmbH, Taufkirchen, Germany), respectively. To establish a negative control, slices were incubated with a DMSO-containing culture medium in a manner consistent with its use in siponimod- or fingolimod-treated slices (Figure 4). For optimal substance penetration into the hippocampal tissue, 100 μL of the respective substance-enriched medium was applied on top of the membrane inserts onto the hippocampal slices. The timing of substance application corresponded to the imaging time point of the same condition on the following day.

### 4.4. Live Imaging

Before mitochondrial imaging on day 15, the membranes were washed with fresh, pre-warmed culture medium and cut out of the cell culture insert using a sterile scalpel (Figure 5A,B). Membranes were transferred into a 6-well glass-bottom plate (P06-1.5H-N, CellVis, Mountain View, CA, USA) containing pre-warmed imaging medium (1× FluoroBrite DMEM (Gibco^TM^, Thermo Fisher Scientific Inc., Waltham, MA, USA), 1% HEPES, 1% Pen/Strep) (Figure 5C), with the hippocampal slices facing the glass bottom. A small handmade grid (stainless steel) was gently placed on top of the membrane (Figure 5D) to prevent movement during the imaging procedure without manipulation of the slices.

For mitochondrial live imaging, we used a Nikon Spinning Disk Confocal CSU-X (Nikon Corporation, Minato, Tokyo, Japan). The conditions of the integrated incubation chamber were set to 37 °C and 5% CO_2_ (Figure 5E) and the membrane-containing glass-bottom plate was transferred into the dedicated imaging chamber (Figure 5F). For an initial overview and identification of the DG as an area of particular interest for live imaging, the ocular mode was used at 40× magnification with 30% laser power and 300 ms exposure time (40× Apo Water Objective, NA 1.25, WD 180 µm). The DG represents a region well-explored in previous studies on neurodegeneration and brain damage [89] because it shows excellent neuronal long-term survival in chronic slice cultures [52]. The DG occupies a more protected and deeply embedded position within the slice, ensuring its preservation within the culture, while the CA1–3 (cornu ammonis) regions, positioned along the slice edges, are more exposed and, therefore, more susceptible to damage [53]. Per culture condition, four slices with adequately maintained DGs were identified (Figure 5G). Subsequently, within these DGs, three random regions of interest (ROI) were imaged, respectively, resulting in a total of twelve time-lapse sequences per treatment condition (12 ROIs). Time lapses of 2 min were acquired in the confocal spinning disc mode, generating an image every four seconds (31 images per sequence) (Figure 5H). The CFP fluorophore located within the neuronal mitochondria was excited at 488 nm wavelength and emission was detected between 525 nm and 550 nm. Due to continuous photobleaching, the laser power had to be adapted during image acquisition (max 40%). The imaging protocol was designed to have a maximum duration of 60 min to guarantee optimal slice survival throughout the entire imaging procedure.

### 4.5. Statistics

The data were statistically analyzed using GraphPad Prism 8.4.3 (GraphPad, CA, USA). Group comparisons were performed by Kruskal–Wallis test, followed by Dunn’s post hoc test. *p* values < 0.05 were considered significant. The significance of the data was further depicted as * implying *p* < 0.05, ** implying *p* < 0.01, *** implying *p* < 0.001, and **** implying *p* < 0.0001.

## 5. Conclusions

Our ex vivo results confirm previous findings on oxidative stress-induced morphological mitochondrial alterations [47,48]. Oxidatively induced changes in mitochondrial morphology can be partly prevented by siponimod at 1 nM, while a similar trend is seen for fingolimod. Finally, oxidative stress-induced alterations in mitochondrial motility are reduced by both 1 nM siponimod and fingolimod.

## Figures and Tables

**Figure 1 ijms-25-00261-f001:**
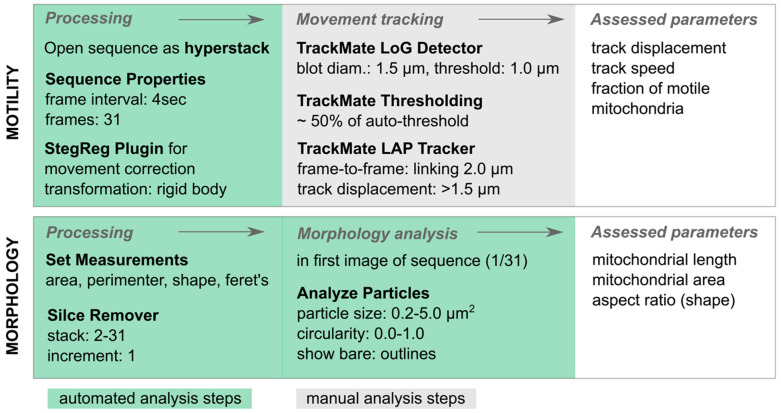
Overview of semi-automated analysis steps. Green background highlights automated steps of the semi-automated in-house developed macro, while grey identifies manual steps. The mitochondrial motility is assessed in 2 min sequences with 31 images each (1 image every 4 s) using TrackMate. For mitochondrial morphology measurements, particles are analyzed in the first image of the sequence after removal of images 2–31.

**Figure 2 ijms-25-00261-f002:**
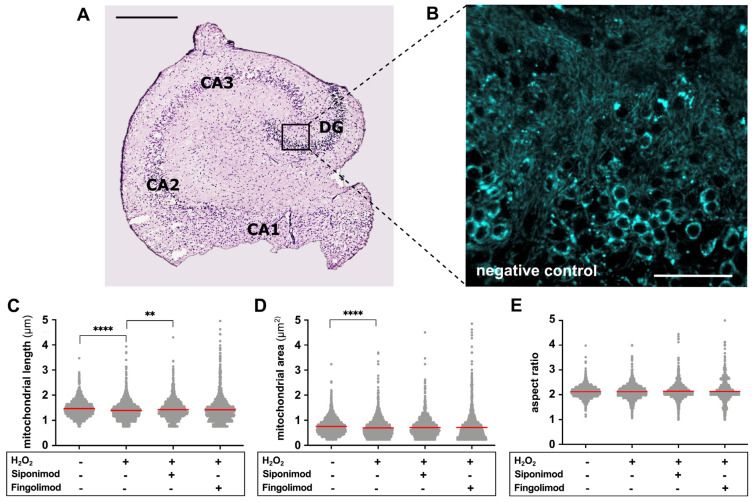
Analysis of mitochondrial area, length, and aspect ratio. (**A**) Exemplary H&E stained hippocampal section (brightfield microscopy) indicating the DG as the region of interest for live imaging as well as the subfields of cornu ammonis (CA1, CA2, and CA3). Scale bar: 100 µm. (**B**) Exemplary DG fluorescence image of the negative control condition in which morphological analysis is performed (40× Apo Water Objective). Scale bar: 50 µm. (**C**–**E**) Results of the semi-automated analysis of mitochondrial length, area, and aspect ratio comparing H_2_O_2_ (*n* = 16803) vs. negative control (*n* = 16569), H_2_O_2_ vs. H_2_O_2_ + siponimod (*n* = 17709), and H_2_O_2_ vs. H_2_O_2_ + fingolimod (*n* = 15161) using the Kruskal–Wallis test with preselected pairs of columns, followed by Dunn’s post hoc test. Red lines indicate the mean. ** implies *p* < 0.01 and **** implies *p* < 0.0001.

**Figure 3 ijms-25-00261-f003:**
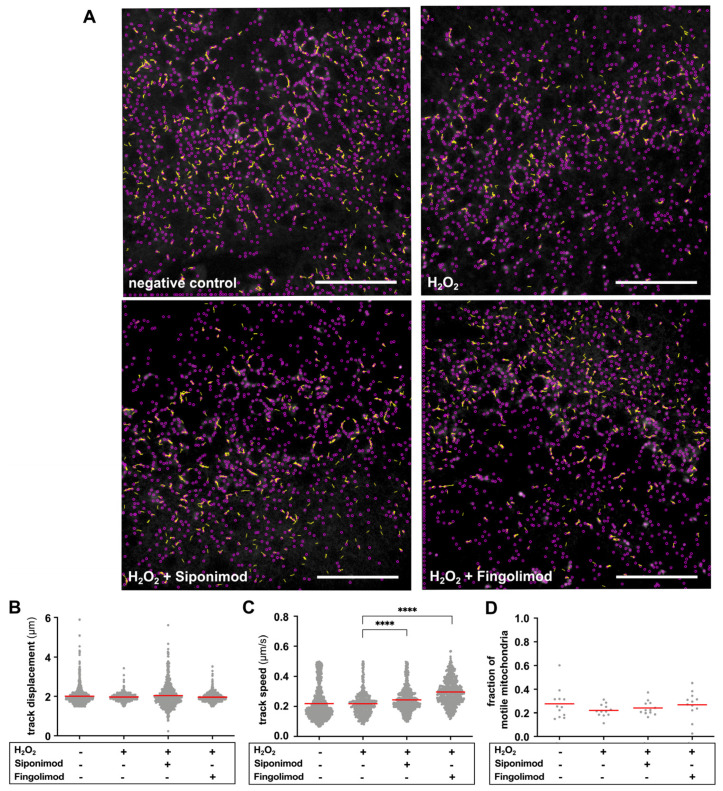
Analysis of mitochondrial track displacement, speed, and fraction of motile mitochondria. (**A**) Exemplary TrackMate images of the four different culture conditions highlight all detected mitochondria with purple dots; mitochondrial movement is indicated with yellow lines (length of lines indicates track displacement; >1.5 µm was considered). The fraction of motile mitochondria decreased upon oxidative stress and approached control fractions upon treatment with H_2_O_2_ + siponimod or fingolimod (fractions of ROIs: neg. control = 0.319; H_2_O_2_ = 0.210; H_2_O_2_ + siponimod = 0.307; H_2_O_2_ + fingolimod = 0.311). Scale bar: 50 µm. (**B**–**D**) Results of the semi-automated analysis of mitochondrial track displacement and speed and the fraction of motile mitochondria comparing H_2_O_2_ (*n* = 4693) vs. negative control (*n* = 3600), H_2_O_2_ vs. H_2_O_2_ + siponimod (*n* = 4328), and H_2_O_2_ vs. H_2_O_2_ + fingolimod (*n* = 4325) using the Kruskal–Wallis test with preselected pairs of columns, followed by Dunn’s post hoc test. Red lines indicate the mean. **** implies *p* < 0.0001.

**Figure 4 ijms-25-00261-f004:**
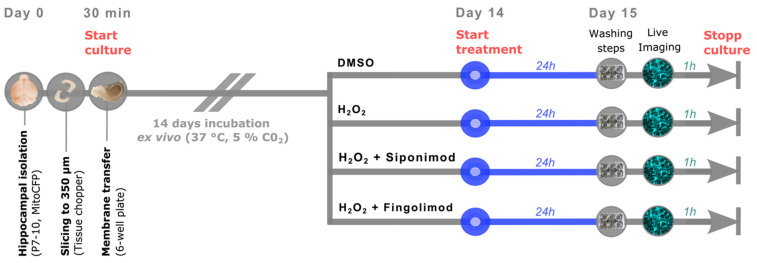
Schematic illustration of the ex vivo study design. On day 0, *mitoCFP* pups aged 7 to 10 days were sacrificed and hippocampi were isolated. Using a tissue chopper, hippocampal sections were generated and transferred to culture inserts. After 14 days of naïve incubation ex vivo, slice treatment was initiated for 24 h using four distinct culture conditions: (1) DMSO (negative control), (2) H_2_O_2_ (positive control), (3) H_2_O_2_ + Siponimod, and (4) H_2_O_2_ + Fingolimod. On day 15, live imaging of mitochondrial dynamics was conducted for 1 h.

**Figure 5 ijms-25-00261-f005:**
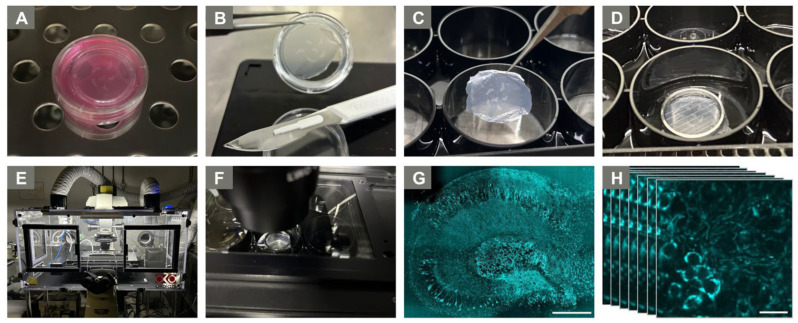
Live imaging set-up. (**A**) Washing of membrane in culture medium (2 × 10 min, 37 °C, 5% CO_2_). (**B**) Cutting out of membrane from culture insert. (**C**) Membrane transfer to glass-bottom plate with imaging medium. (**D**) Grid placement to prevent membrane movement. (**E**) Incubation chamber of the Nikon Spinning Disk Confocal CSU-X. (**F**) The glass-bottom plate inside the imaging chamber. (**G**) Exemplary CFP fluorescence image of the well-preserved hippocampal slice (stitched, 40× Apo Water Objective). Scale bar: 100 µm. (**H**) 2 min time-lapse acquisition (DG region, stack of 31 images, 40× Apo Water Objective). Scale bar: 20 µm.

**Table 1 ijms-25-00261-t001:** Summary of results on mitochondrial morphology.

	Mitochondrial Length (µm)	Mitochondrial Area (µm^2^)	Aspect Ratio (1–∞).
	*Mean ± SD*	*Median*		*KW*	*Mean ± SD*	*Median*		*KW*	*Mean ± SD*	*Median*		*KW*
**Negative control**	1.469 ± 0.317	1.443	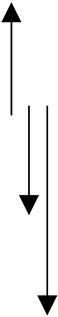	****	0.755 ± 0.324	0.712	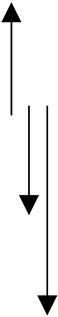	****	2.118 ± 0.228	2.108	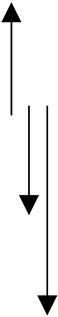	>0.1
**H_2_O_2_**	1.400 ± 0.353	1.370		0.698 ± 0.371	0.630		2.119 ± 0.280	2.110	
**H_2_O_2_ + Siponimod**	1.433 ± 0.362	1.398	**	0.711 ± 0.358	0.665	>0.1	2.140 ± 0.306	2.106	>0.1
**H_2_O_2_ + Fingolimod**	1.426 ± 0.493	1.368	>0.1	0.713 ± 0.509	0.607	>0.1	2.130 ± 0.410	2.076	>0.1

** *p* < 0.01, **** *p* < 0.0001; SD, standard deviation; KW, Kruskal–Wallis test.

**Table 2 ijms-25-00261-t002:** Summary of results on mitochondrial motility.

	Track Displacement (µm)	Track Speed (µm/s)	Fraction of Motile Mitoch.
	*Mean ± SD*	*Median*		*KW*	*Mean ± SD*	*Median*		*KW*	*Mean ± SD*	*Median*		*KW*
**Negative control**	2.009 ± 0.416	1.935	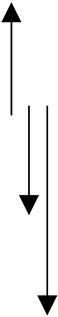	>0.1	0.219 ± 0.107	0.186	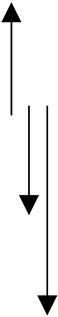	>0.1	0.277 ± 0.128	0.252	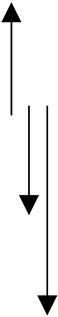	>0.1
**H_2_O_2_**	1.968 ± 0.197	1.956		0.219 ± 0.081	0.209		0.220 ± 0.054	0.212	
**H_2_O_2_ + Siponimod**	2.037 ± 0.628	1.938	>0.1	0.244 ± 0.081	0.231	****	0.241 ± 0.059	0.223	>0.1
**H_2_O_2_ + Fingolimod**	1.962 ± 0.209	1.931	>0.1	0.296 ± 0.084	0.289	****	0.269 ± 0.087	0.284	<0.1

**** *p* < 0.0001; SD, standard deviation; KW, Kruskal–Wallis test.

## Data Availability

The data analyzed in the present study are available from the corresponding author upon reasonable request.

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
