# Peer review of "Investigating the Mitoprotective Effects of S1P Receptor Modulators Ex Vivo Using a Novel Semi-Automated Live Imaging Set-Up"

_ijms, 2023, doi:10.3390/ijms25010261_

Round 1
Reviewer 1 Report
Comments and Suggestions for Authors
Mitochondria play many important roles in cell biology, and mitochondrial dysfunction contributes to a variety of disorders, including neurodegenerative conditions and, potentially, multiple sclerosis. Thus, assays for mitochondria in living neurons are very helpful for understanding cell biology as well as testing drug candidates. Here, Ludwig and colleagues establish, to the best of my knowledge, one of the first studies for using intact chronic hippocampal slices to monitor mitochondrial alterations in morphology and motility within living brain tissue, which the authors then use to investigate the effects of hydrogen peroxide as well as the S1P receptor inhibitors, Fingolimod and Siponimod. An assay of this type would be very helpful for researchers in many labs. While the story is relatively straight forward, I found a few points a little confusing.
The central hypothesis centers on a connection between the S1P receptor and mitochondria. However, I cannot seem to follow the logic. The S1P receptor is, to my knowledge, not located on mitochondria, nor have there been mitochondrial functions linked directly to S1P receptor activity. This hypothesis seems to rest solely on one citation (reference 30), which also doesn’t seem to make a clear link between the S1 receptor and mitochondrial activity (instead showing that antioxidant pathways such as Nrf2 and glutathione are up regulated). It would be very helpful if the authors could better clarify the possible molecular mechanistic link between the S1P receptor and mitochondria.
On a technical point, I have several questions about the stressor, which, in this study, was hydrogen peroxide (H2O2). However, H2O2 shows general toxicity through a variety of potential mechanisms. Thus, H2O2 is not specific to mitochondria, making me wonder why it was selected. The citation cited by the authors used menadione, which directly generates reactive oxygen species via mitochondria. Why did the authors opt for a different stressor? Can that please be described in more detail?
On a related note, I also wonder about how the concentration of H2O2 was also selected. 24 hours with even moderate concentrations of H2O2 can be toxic to many cells. How did the authors choose the duration as well as the concentration being used in this study?
The authors invested considerable effort into image analysis. However, the reporting of this could be improved. It would help both figures 2 and 3, if the raw images as well as the segmented images could be shown side by side. One reason is that the segmented mitochondria in figure 3 seem to show only small round mitochondria (purple circles). However, in figure 2, I can visualize several elongated mitochondria using the mito-CFP signal. Thus, I wonder about the efficiency of the segmentation. It would help to see the pre- and post-segmented images side by side with a detailed description by the authors on this issue.
The graphs in figures 2 and 3 involve very small differences within a dataset that is quite heterogeneous, making it hard to visualize the changes in the mean. Maybe it would help to use a y-axis that is log base 2 instead of linear?
The legends for figures 2 and 3 should describe how many biological replicates were performed.
Reviewer 2 Report
Comments and Suggestions for Authors
This study investigated the effects of siponimod and fingolimod on neuronal mitochondria under oxidative stress, using hippocampal slice cultures from mitoCFP mice exposed to hydrogen peroxide (H2O2). The research focused on changes in neuroaxonal mitochondria within the dentate gyrus (DG) and compared siponimod's effects with those of fingolimod, its predecessor.
H2O2 treatment significantly reduced mitochondrial length and area in hippocampal slices. Siponimod treatment for 24 hours mitigated these oxidative stress-induced changes, showing a significant increase in mitochondrial length and a trend towards larger mitochondrial areas compared to H2O2-only treatment. Fingolimod showed a trend towards improvement but was not significantly effective. The mitochondrial aspect ratio remained unchanged across all treatment conditions. Beyond, oxidative stress led to a non-significant reduction in the distance mitochondria traveled and no effect on their speed. However, the fraction of motile mitochondria decreased significantly. Siponimod showed a tendency to prevent the decrease in mitochondrial displacement. Both siponimod and fingolimod significantly increased mitochondrial track speed compared to H2O2 treatment alone, with motility fractions approaching control levels. Fingolimod treatment came close to statistical significance in increasing the fraction of motile mitochondria compared to H2O2 alone. The presented results indicate that siponimod and, to a lesser extent, fingolimod can mitigate some of the detrimental effects of oxidative stress on mitochondrial morphology and motility in hippocampal neurons.
While the presented findings are novel and interesting some improvements would strengthen the manuscript.
1. The authors state that „SPMS can be targeted with a few specific DMTs, namely siponimod (1st line therapy), ozanimod, and mitoxantrone”. Please complete the list listing all approved drugs for SPMS
2. The authors state that “Importantly, siponimod 80 does not modulate S1PR3, which is located on cardiac cells and is thought to be responsible for cardiac complications observed for fingolimod, such as bradycardia and arrhythmia”. Please consider that this might well be true for mice but not humans, where cardiac monitoring is required as well in siponimod-treated patients.
3. The recent literature, demonstrating protective effects of siponimod should be cited.
4. It was not clear to me how the authors can conclude that specifically axonal mitochondria were analyzed. Please specify.
5. I was wondering whether in the applied model, glia activation occurs as a response of the H2O2 treatment. This could be analyzed by IF or gene expression studies.
6. Please state why the DG was in the focus of the study.
7. Since the presented work has a strong methodological aspect I strongly suggest to present all key methods in the main paper and not in the supplements.
8. I was wondering whether the protective effects of siponimod and/oe fingolimod might have been more pronounced in case the slices would have been pre-treated with the compounds.
Round 2
Reviewer 2 Report
Comments and Suggestions for Authors
I have carefully reviewed the revised version of the manuscript submitted for publication. I appreciate the effort put into revising the manuscript; however, I must express my disappointment regarding specific aspects that have not been addressed as suggested.
One of the main concerns in my previous review was the lack of experimental investigation into glial activation and pre-treatment with S1PR modulators. These experiments were critical in providing a more comprehensive understanding of the subject matter. Unfortunately, it appears these suggested experiments have not been conducted in the revised manuscript. These experiments would have not only strengthened the manuscript's scientific rigor but also enhanced its appeal to a broader audience.
I understand that conducting additional experiments might require substantial time and resources, but I believe the potential impact on the manuscript's quality and the advancement of knowledge in this area would be significant. I look forward to seeing a more comprehensive version of your work that addresses these critical aspects.
Round 3
Reviewer 2 Report
Comments and Suggestions for Authors
The authors have adressed all the points raised